# Overcoming extrapolation challenges of deep learning by incorporating physics in protein sequence-function modeling

Shrishti Barethiya[1], Jian Huang[1], Clarice Stumpf[1], Xiao Liu[2], Hui Guan[2], Jianhan Chen [1]*

**1** Department of Chemistry, University of Massachusetts, Amherst, Massachusetts, United States of America, **2** College of Information and Computer Sciences, University of Massachusetts, Amherst, Massachusetts, United States of America

* jianhanc@umass.edu

## Abstract

Understanding protein sequence-to-function relationship is crucial to assist studies of genetic diseases, protein evolution, and protein engineering. The sequence-to-function relationship of proteins is inherently complex due to multi-site high-dimensional correlation and structural dynamics. Deep learning algorithms such as (graph) convolutional neural networks and recently transformers have become very popular for learning the protein sequence-to-function mapping from deep mutational scanning data and available structures. However, it remains very challenging for these models to achieve accurate extrapolation when predicting functional effect of variants with positions or mutation types not seen in the training data. We propose that incorporating the physics of protein interactions and dynamics can be an effective approach to overcome the extrapolation limitations. Specifically, we demonstrate that biophysics-based modeling can be used to quantify the energetic effects of mutations and that incorporating these physical energetics directly within the convolution and graph convolution neural networks can significantly improve the performance of positional and mutational extrapolation compared to models without biophysics-inspired features. Our results support the effectiveness of leveraging physical knowledge in overcoming the limitation of data scarcity.

## Author summary

Deep learning has fundamentally transformed science and research in recent years. Yet, many problems in biophysics and biochemistry remain inaccessible to traditional deep learning due to a lack of large training data. Incorporating physical principles in machine learning is arguably required to overcome data scarcity. In this work, we examine the effectiveness of incorporating biophysics-based

**Data availability statement:** All training data and scripts are available on GitHub (https://github.com/SBarethiya/VEP).

**Funding:** This work is supported by NIH R35 GM144045 (to JC). The funders had no role in study design, data collection and analysis, decision to publish, or preparation of the manuscript.

**Competing interests:** The authors have declared that no competing interests exist.

features in deriving more reliable predictors of the effects of sequence variants on protein function. Our results show that including the energetics of mutational effect on protein stability can significantly improve machine learning models' ability to predict novel mutations not seen in the training data set, especially for mutations on novel sequence positions. Further incorporation of sequence evolutionary information offered by pre-trained protein large language models could further improve the predictive power. Our work thus provides an efficient framework for training better variant effect predictors from deep mutational scanning dataset. The result predictors can aid protein engineering and the prioritization of studying genetic variations in diseases.

## Introduction

Proteins play crucial and diverse roles in nearly all biological processes. The function of a specific protein is ultimately dictated by its amino acid sequence. The physico-chemical properties of amino acids, their spatial arrangements as defined by three-dimensional structures, and conformational dynamics together provide the physical basis of protein biological functions. Understanding the protein sequence-to-function relationship, that is, how sequence governs their biological functions, is a fundamental problem central to the studies of protein evolution, diagnosis of genetic disease, and protein engineering and design [1–4]. While homology-based annotation of protein function given the sequence has been highly successful [5–7], the sequence-to-function mapping remains inherently challenging. This is especially true in the prediction of functional effects of one or a few specific mutations of a given protein, due to the complex high-dimensional correlation between sequence and function that arise from frequently nontrivial impacts on physiochemical, structural and dynamic properties of the protein.

In the past couple of decades, advancements in high-throughput functional assays, especially deep mutational scanning or DMS [8–12], have generated large sets of functional readouts in the sequence space of protein variants. DMS integrates large genetic mutant libraries, high-throughput phenotyping, and DNA sequencing of both input and post-selection cells, enabling examination of functional impact of up to tens of thousands of mutants in a systematic and economical fashion [13–15]. Each step in a DMS experiment may have inherent limitations. For example, the library construction technique can introduce mutation bias, off-target issues, or efficiency constraints, and, most critically, it is typically limited to handling up to tens of thousands of variants [16,17]. Since the sequence space grows exponentially with length, achieving comprehensive DMS analyses of large proteins becomes prohibitively expensive and is generally infeasible. Leveraging data from DMS experiments, genome-wide sequencing [18], and vast computational and experimental structural databases [19–22], machine learning (ML) models have been trained to make quantitative predictions of sequence-to-function molecular phenotypes [23–29]. These predictors, also commonly referred to as variant effect predictors (VEPs), provide

powerful tools for many applications, including designing a protein variant optimizing functions of naturally occurring proteins [30], creating protein variants of exceptional fitness [31,32], and developing treatment strategies of genetic diseases [33–36].

VEPs can be broadly categorized into unsupervised and supervised learning models. These models generally utilize a combination of four major input feature types: protein wildtype or mutant sequence, amino acid properties, protein structure, and evolutionary information [27]. Unsupervised learning models typically exploit extensive protein sequence data and multi sequence alignment (MSA) to learn complex epistatic interactions or constraints between sites. For example, DeepSequence [37] is a Bayesian deep latent-variable model trained on large DMS datasets of proteins and RNA domains. This model learns the probability distribution of sequences, predicts mutational effects, and models the evolutionary fitness landscape within families. SeqDesign [38] is an alignment-free autoregressive generative model that uses dilated convolutional neural networks trained on protein family sequences, achieving similar prediction accuracies benchmarked on 40 protein DMS datasets. SeqDesign offers major improvements where MSA is not robust, such as predicting effects of indels, disordered proteins, and designing proteins like antibodies.

Supervised learning VEPs such as Envision [25], positive-unlabeled learning [39], and ECNet [40] are trained on experimental mutagenesis/DMS datasets, using approaches ranging from decision trees to protein language models. In particular, Gelman et al [41] have designed the NN4dms deep learning framework that includes a set of amino acid properties (AAIndex [42]) features and/or 3D structures in addition to sequences, and evaluated four ML algorithms ranging from linear regression (LR), fully connected neural network (NN), sequence convolutional neural network (CNN), to graph convolutional neural network (GCN). All four models performed well with DMS datasets containing single and higher-order mutations compared to biophysics-based and unsupervised methods, especially with CNN and GCN. A key lesson is that the performance depends critically on the sequence coverage. This is particularly evident when challenged with mutational and positional extrapolations [41,43]. In mutational extrapolation, models aim to predict the effects of mutation types not observed at the site of interest during training; in positional extrapolation, models aim to predict the effects of mutations at entirely unseen sites. All NN4dms models struggle in mutational extrapolation and largely fail in positional extrapolation in all testing cases [41]. These results highlight arguably the most common bottleneck in training supervised VEPs, the data scarcity problem, especially for larger proteins where comprehensive coverage in DMS experiments becomes infeasible.

Novel information must be included to supplement DMS datasets to overcome the data scarcity problem in general. One elegant strategy is to use pre-trained models, such as large language models (LLMs) derived from sequence databases. For example, the geometric vector perceptron multi-sequence alignment (GVP-MSA) model [32] utilizes pre-trained MSA transformer model [44] to encode higher-order sequence dependencies, which is then integrated into the GVP-graph neural network alongside extracted structural features. The model has shown to perform well for positional extrapolation of single-variant effects and predicting the higher-order mutational effect from datasets containing only single mutations. Another strategy is to incorporate fundamental biophysics of protein structure, dynamics and interactions during model training [45–47]. The premise here is that these physical principles ultimately determine how any sequence variation may affect function and are thus ideally suited for overcoming extrapolation limitations. In a recent mutational effect transfer learning (METL) model [48], a transformer-based neural network was pre-trained on large-scale synthetic data derived from Rosetta-based molecular modeling [49] and then fine-tuned on the experimental DMS data. The authors evaluated two types of models, METL-local and METL-global, the first of which is protein-specific and the latter is trained on a broad range of proteins [48]. Even though METL-global suffers from severe over-fitting, both models demonstrated superior performance compared to NN4dms when sequence coverage is limited. Especially for positional extrapolation, modest Spearman correlation between ~0.20 - 0.75 was achieved for all test proteins with either METL-local or METL-global [48]. These successes clearly support the effectiveness of incorporating biophysics to overcome data scarcity in deep learning.

In this work, we developed an efficient and direct strategy for incorporating biophysics-based features to train accurate supervised protein-specific VEPs and to overcome both mutational and positional extrapolation challenges. The strategy builds upon our previous work in predicting the mutational effects of the gating voltage of big potassium (BK) channels [50], where random forest models, trained with less than 500 data points, achieve accurate performance when supplemented with biophysical features including Rosetta-based energy terms and structural dynamics from all-atom simulations. The predictor correctly recapitulates the hydrophobic gating mechanism of BK channels and captures nontrivial voltage gating properties in regions where few mutations are known [50]. Several novel mutations were subsequently confirmed by experiments. The workflow presented in this work only requires evaluation of Rosetta energies for all 19 possible single mutations for each residue site of a given protein, each of which involves a short optimization of sidechain repacking within the vicinity of the mutation. The resulting Rosetta energy terms are used directly to quantify the impacts of any single or multiple variants on various types of physical interactions, assuming additivity. The resulting CNN and GCN models are comparable to those trained without biophysical features in NN4dms for random splitting but provide superior extrapolation performance, which is comparable to METL. The overall workflow is highly efficient, mainly requiring only sequence length x 19 Rosetta rotamer repacking evaluations, and can be readily extended to any protein or protein complex, allowing an accurate VEP trained within few minutes to hours given the DMS dataset.

## Methods

### Model proteins and DMS datasets

Following NN4dms [41], five protein DMS datasets are considered in this work, including avGFP (*Aequorea victoria* green fluorescent protein) [51], Bgl3 (*β*-glucosidase enzyme) [11], GB1 (IgG-binding domain of protein G) [52], Pab1 (RRM2 domain of the *Saccharomyces cerevisiae* poly(A)-binding protein) [13], and Ube4b (U-box of the murine E3 ligase ubiquitination factor E4B) [53]. These proteins cover sizes from 56 to 501 residues and represent distinct topologies from helical, mixed helix/beta to full beta-barrel (**Fig 1**). Their sequence lengths, functions, and DMS dataset sizes are summarized in Table 1. The functional scores are available publicly for the avGFP, Pab1, and Ube4b, whereas for GB1 and Bgl3, functional scores are processed through Enrich2 [54] by Gitter and Coworkers [41] based on raw sequencing read counts. The dataset consists of substitutions and deletion mutations. Before the model training, all the deletions have been removed from the datasets and the final datasets only contain missense substitutions. The distributions of final DMS data points (Fig 1) display signficant heterogeneity along the protein sequences except for GB1 with exceptional number of over half million variants. Most variants in the datasets are single and double mutation variants (S1A Fig), with avGFP containing the broadest distribution in number of mutations per variants (up to 15). The higher order variants could inform correlation between different mutation sites and thus potentially benefit VEP models to make more accurate positional and mutational extrapolations. Of note, only 43% and 38% of possible site mutations are represented for larger proteins avGFP and Bgl3, respectively, and many of mutations are only sample as part of high order multi-site mutation variants (S1B Fig). Heterogenity of structures, functions, DMS data distributions of these five protein cases provides a good benchmark for model performance assessments.

### Calculation of Rosetta energetic features

The Rosetta FastRelax protocol [55] was used to relax the initial PDB structure of each protein for 5 independent runs of 100 iterations. During relaxation, both backbone and sidechain degrees of freedom were allowed, and the lowest energy pose from 5 runs was eventually selected as the starting point for the subsequent mutational ΔΔG calculations.

The predict_ddG.py script from the PyRosetta [56] package was used to calculate the free energies of all possible single mutations of each protein. Within the script, sidechain substitution is first made at a given site, and the PackRotamerMover method is then used to repack sidechains of surrounding residues within a certain cut-off radius using the

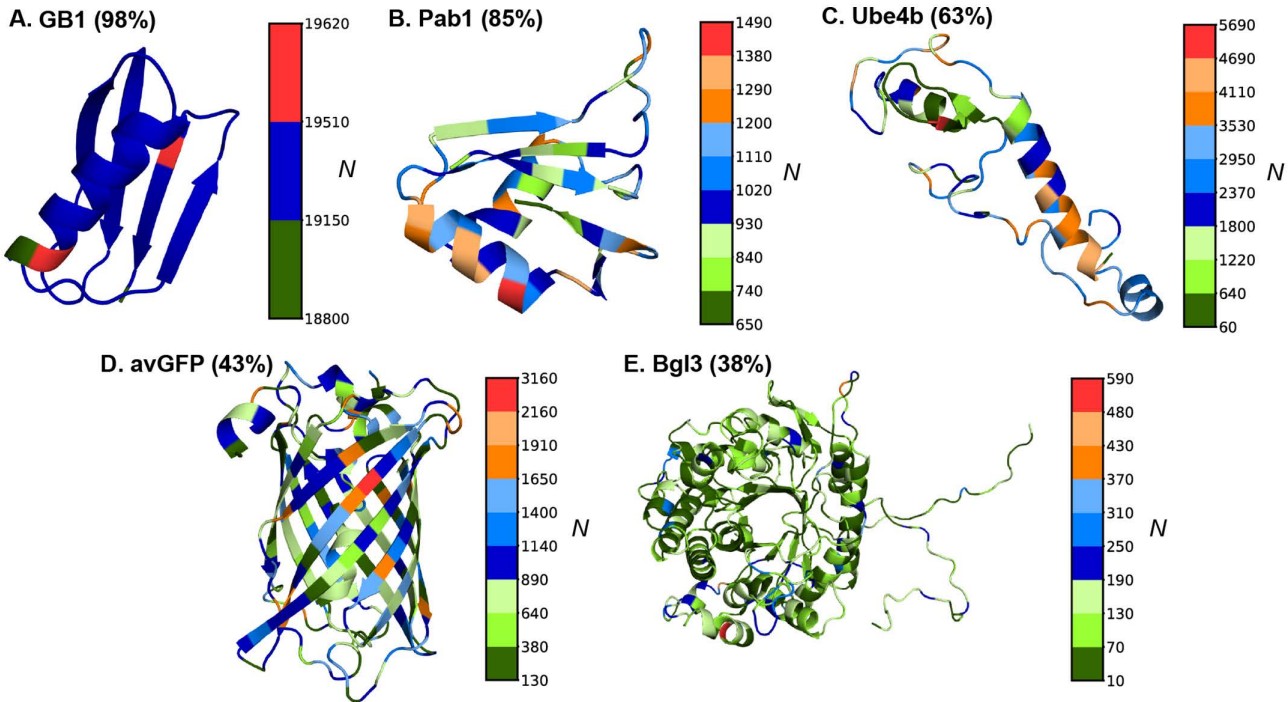

**Fig 1. Model protein structures and mutational data point distributions.** All proteins are represented using cartoons with PDB structures. The PDB IDs used are: 2QMT (GB1), 1CVJ (Pab1), 2KR4 (Ube4b), 1EMM (avGFP), and 1GNX (Bgl3). Each site on the structure is colored based on the number of DMS variants containing mutations at that site **(N)**. The coverage of unique mutations sampled in the dataset is given in parenthesis next to the protein name.

**Table 1. Summary of the five proteins and DMS datasets, ordered in decreasing percentage of sequence space coverage in the dataset.**

| Protein | length | Function | Total data points | Substitution data points | Mutational extrapolation | Positional extrapolation |
|---------|--------|----------|-------------------|--------------------------|--------------------------|--------------------------|
| **GB1 [52]** | 56 | Immunoglobin G-binding | 536,084 | 536,084 | ~311,400 | ~305,000 |
| **Pab1 [13]** | 75 | Poly (A)-binding | 40,852 | 37,710 | ~22,300 | ~21,800 |
| **Ube4b [53]** | 102 | Auto-ubiquitination activity | 98,297 | 91,031 | ~44,000 | ~43,400 |
| **avGFP [51]** | 237 | Fluorescence | 54, 024 | 51,714 | ~17,600 | ~17,300 |
| **Bgl3 [ 11]** | 501 | Hydrolysis of beta-D-glycosyl residue | 26,653 | 25,737 | ~16,001 | ~15,946 |

Dunbrack rotamer libraries [57] and minimize the conformational energy based on the REF2015 energy function [49]. For an input structure, the script will internally calculate the ΔG values (the folding Gibbs free energy versus the fully unfold status) for the wildtype and a mutant with a specified mutation site after repacking and report the final ΔΔG (= $\Delta G_{wildtype}$ -$\Delta G_{mutant}$). To achieve stable energy evaluations and balance the tradeoff between re-packing iterations and computational cost (few seconds per mutation), we focused on optimizing two key parameters for the protocol (see detailed discussion in the Results section): the cutoff radius (repacking radius) and *nloops*. The repacking radius is used to select residues to be repacked by including surrounding residues that have a $C_\alpha$ (for glycine) or $C_\beta$ (other residues) atoms within the cutoff radius of the mutated site. The *nloop* parameter defines the sidechain repacking iterations before outputting the final score.

We have shown that it is advantageous include individual Rosetta energy components in addition to the total ΔΔG [49]. Even though the Rosetta energy score function consists of 19 energy terms, only 8 terms with substantial variances were selected to be included in training: Lennard-Jones attractive and repulsive between atoms in the same and different residue, Lazaridis-Karplus solvation energy, Coulombic electrostatic potential with distance-dependent dielectric, backbone hydrogen bonds, sidechain-backbone and sidechain-sidechain hydrogen bond energy, disulfide geometry potential, energy term based on Ramachandran maps, probability of amino acid at different backbone torsion and difference in Rosetta free energy of the wildtype (S1 Table).

## Atomistic molecular dynamics simulations

Short atomistic molecular dynamics (MD) simulations with explicit solvents and ions were conducted for each protein to quantify the inherent flexibility of each residue in the context of the protein structure and its native-like environment, as characterized by the root mean square fluctuation (RMSF) of the $C_\alpha$ atom. The initial PDB structures (**Fig 1**) were first cleaned to preserve only the protein part and then solvated with TIP3P water molecules and 0.15 M KCl using CHARMM-GUI [58,59]. The CHARMM36 force field [60] was used, and all simulations were conducted using the GPU-accelerated GROMACS 2019 [61]. The non-bonded forces were calculated using a cut-off of 12 Å and a smoothing switching function to attenuate forces from 10 Å. The particle mesh Ewald (PME) [62] algorithm was employed for long-range electrostatic interactions. All hydrogen-containing bonds were constrained with the LINCS [63]. For equilibration, all systems were first energy minimized for 5000 steps using the steepest descent algorithm and then were subjected to another 125000 steps of restrained NVT (constant particles, constant volume, and constant temperature) simulations with 1 fs as the integration timestep, where the protein backbone and sidechains were positionally restrained with harmonic force constants of 400 kJ/mol/Å$^2$ and 40 kJ/mol/Å$^2$, respectively. The final unrestrained production MD simulations were performed for each system for 200 ns with leap-frog integrator and time step of 2 fs with Nose-Hoover thermostat [64,65] at 303.15 K and Parrinello-Rahman barostat [66] at 1.0 bar and compressibility of 4.5x10$^{-5}$ bar. Snapshots were saved every 100 ps. To calculate the residue RMSF profile, all frames were first aligned with respect to the first frame using backbone atoms and then RMSF values were then calculated for all $C_\alpha$ atoms (S2 Fig). The MD simulations are intended to roughly capture the backbone flexibilities of each residue site of the wild-type proteins, and thus short simulations are sufficient for this purpose. We note that some mutations could lead to nontrivial effects on protein structure and dynamics and that the effects of these mutations would be extremely difficult to predict and capture in VEPs.

## Input features and preprocessing

Four types of input features were used for model training: one-hot encoding of amino acids, amino acid index (AAIndex), Rosetta energy terms, and RMSF from MD simulations, as summarized in S1 Table. For the graph convolutional network (GCN), the structure of each protein used to initiate MD simulations was used for the construction of the graph. The AAIndex database consists of a set of 566 numerical values describing various physicochemical and biochemical properties for each canonical amino acid [42]. Due to strong correlations among many properties, only 19 principal components (with 100% variance) from PCA analysis of the original AAIndex features [67] were used in the current work after the min-max normalization to the range of [0, 1]. For Rosetta energies (19 components and the total ΔΔG) features, the tanh squashing function [50] was first used to squash energy features greater than 50 REUs (Rosetta energy units), as exceedingly large energy values often arise from steric clashes and are not quantitatively meaningful. Additionally, energy features with small variances were discarded, and only 8 features were used in the final modeling training and testing (S1 Table and S6 Fig). Finally, the min-max normalization was applied to the selected 7 energy features and the total ΔΔG. Features at each position was treated independently, and Rosetta features were assigned to the corresponding mutation within each variant without assuming additivity or cooperativity. The $C_\alpha$ RMSF values from MD simulations were similarly min-max normalized.

Lastly, sequence positional information using one-hot encoding was expressed as a binary vector of length 21 (20 for natural amino acid and the extra one for the stop codon).

To further augment biophysics-based features with sequence evolution information, the log-likelihood ratio (LLR) scores of all site-specific mutations was computed using the masked-marginal scoring function [68] for four ESM-2 models with 8M, 35M, 150M, and 650M parameters [22]. The LLR score quantifies the probability of observing a specific residue type at the given site of the sequence based inferred from sequence evolution. The raw LLR score was used in all extrapolation experiments.

### ML model design, training and evaluation

We examined the effects of including Rosetta and MD-derived physical features in the four machine learning algorithms previously evaluated in the NN4dms study [41], namely LR, NN, CNN, and GCN. The leaky ReLU activation function was used in NN, CNN, and GCN, and a dropout rate of 0.2 after the dense layer was used to prevent overfitting. In CNN, the strides were [1] and the padding was valid. For the implementation, python 3.6 and TensorFlow [69] v1.14 were used. In the GCN, the input graph for GCN was constructed based on the distance matrix of $C_\alpha$ (for glycine) and $C_\beta$ (other residues) with a threshold of 6 Å for Bgl3 and 7 Å for other proteins (see the hyperparameter search below). As illustrated in Fig 2, physical features can be directly added to the input layer, which increases the feature dimension from 40 to 49 per residue (8 Rosetta energies + 1 RMSF value) (S1 Table). Similarly in the case of GCN, the Rosetta features were added as node features.

The random splitting of the dataset was done with 70% for training, 15% for validation, and 15% for test. For studies of dataset size dependence, smaller datsets were created by randomly sampling between 100 datapoints and up to 80% of the full dataset. The remaining 20% was consistently used for validation and testing across all experiments, ensuring that the validation and test sets remained the same for every subset. In mutational extrapolation, all unique mutations from the dataset were collected and randomly divided into training (70%), validation (15%), and testing (15%) sets. All variants containing the corresponding mutations were then assigned to the appropriate set. At last all the overlapped variants between the training, testing, and validation were dropped based on unique mutations in the variant. In positional extrapolation, a similar process was used except that splitting was based on positions. In the extrapolation from single mutation variants to higher-order variants, all the single mutation varaints were used for the training (90%) and validation set (10%) as the dataset is limited. All higher-order varaints were used as the testing set, except those for which the corresponding single-mutation variant was not available.

Hyperparameter search was performed on the validation dataset for all models using grid search with different combinations of batch size, learning rate, number of hidden layers, number of nodes, filters and graph threshold. The final selected hyperparameters are reported for the main models as shown in S2 Table. Each model was trained five times with different splits with different random seeds. Pearson and Spearman correlation coefficients were calculated using the SciPy library [70] and mean absolute errors (MAE) were calculated using the scikit-learn [71] between the true and predicted scores.

For the ablation study performed on Pab1 dataset, each architecture was trained by excluding specific biophysics-based feature(s): excluding RSMF, excluding all Rosetta terms, excluding Lennard-Jones terms and ΔΔG, excluding electrostatic terms and ΔΔG, and excluding solvation terms and ΔΔG. To evaluate how the quality of the DMS dataset affects VEP performance, the resampling experiments were performed in a similar manner as described in the NN4dms study [41]. First, the testing set was constructed from randomly selected 10,000 datapoints from the original GB1 dataset. The left-over dataset was used for construction of randomly resampled libraries for training. For each library, multinomial probability distributions were created for both the input and selected sets by normalizing the original read counts of each variant. These distributions provide the probability of generating a read for any given variant. The fraction of reads was computed in both the sets in the base dataset to decide reads sampled from input and selected sets. The sampling of

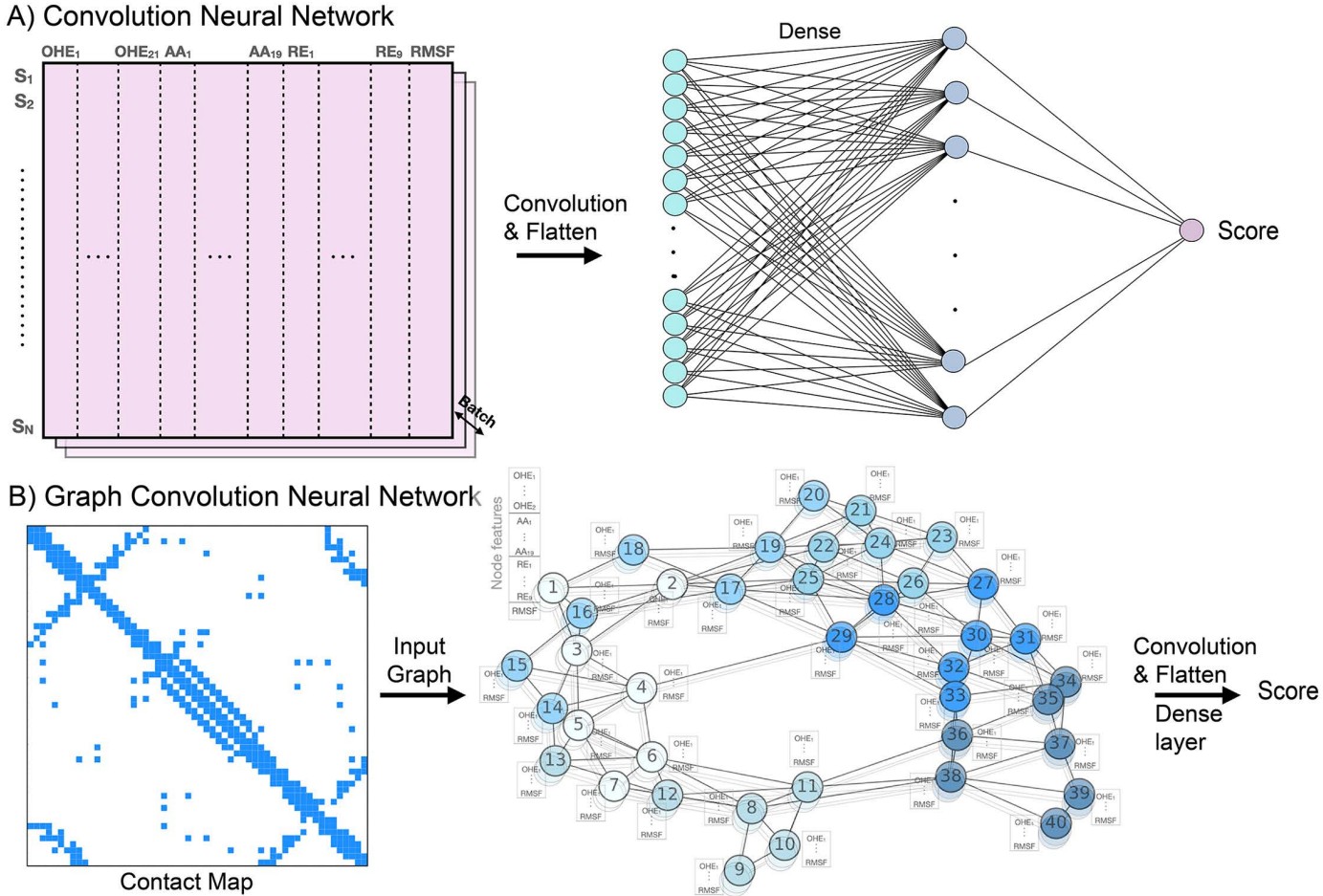

**Fig 2. Machine learning architectures for protein variant effect prediction.** In CNN, the input dimension is $N_{res}$ x 40 without biophysics-based features and $N_{res}$ x 49 with biophysics-based features ($N_{res}$: number of residues). In GCN, the wild-type protein structure was used to derive the contact map as graph, where nodes represent residues and edges represent contacts with certain distance cut-off. $S_i$: the residue index, OHE: one-hot encoding, AA: amino acid index, RE: Rosetta energies, and RSMF: root mean square fluctuations (from MD).

reads was based on this fraction and Enrich2 scores were calculated for each resampled dataset. These datasets were generated five times for each combination of library size and number of reads from the remaining dataset.

## Results

### Efficient evaluation of mutational effects on protein interactions

The balance between preserving structural integrity and certain degrees of flexibility of a protein is crucial for its function. This intricate balance is governed by tertiary arrangements of residues of different physiochemical properties and their interaction networks. Substitutions of residues can reshape this balance, influence the free energy landscape of protein folding and unfolding, modulate the conformational dynamics, and ultimately affect function. Developing an efficient approach for evaluating the impacts of a given mutation on the physical basis of protein function is thus key to developing robust VEP models [72]. The Rosetta software [49], in particular, can be used to estimate ΔΔG of protein folding stability upon mutation [73] and has been extensively validated using several protein mutation datasets [74]. Importantly,

 

the Rosetta all-atom energy function [49] includes both physics-based and empirical terms to allow quantifications of the mutational effects on key local physical interactions such as electrostatics, hydrogen-bonding, van der Waals interactions, and sidechain rotamer states.

Reliable estimation of ΔΔG requires relaxation of the protein conformation in response to mutation, which are generally achieved using either FastRelax or PackRotamerMover protocols in Rosetta. In the FastRelax protocol [55], generally five repeated simulated annealing processes are performed, which consists of re-packing of sidechains within a cutoff radius of a selected site followed by global minimization of all dihedral angles while ramping up the repulsive terms in the energy function. However, in the PackRotamerMover protocol [75], only sidechains within the "repacking" radius of the mutated site are allowed to re-pack and no other degrees of freedom are optimized. FastRelax requires longer computing time up to 10s of minutes, depending on the location of the mutation site and the protein size, whereas PackRotamerMover takes only seconds. Importantly, restricted to local sidechain rotamer resampling in the PackRotamerMover protocol avoids the challenges of consistently capturing non-trivial global structural relaxation, allowing more reliable description of the energetic consequences of a mutation. Furthermore, the VEPs in this work will be trained with individual Rosetta energy components and local minimization of PackRotamerMover provides a more faithful picture of how the mutation affect various types of local interactions. Of note, the input features include residue RMSFs derived from MD simulations, which inform the ability of different sites of the protein to undergo further (backbone) relaxation to accommodate various mutations (see Methods and S2 Fig).

We further optimize the PackRotamerMover protocol to ensure reliable estimation of mutational energetics, focusing on two key parameters: the repacking radius (default: 8 Å) and the number of complete repacking runs before outputting the best energy (default *nloop* = 1). Different repacking radii directly affect the range of residues surrounding the mutated site to be selected and have their sidechain rotamers repacked in the protocol, as illustrated in Fig 3A-3D. The results show that a larger repacking radius of 12 Å is necessary to sufficiently optimize the local sidechain configurations (S3 and S4 Figs) and obtain reliable ΔΔG estimates (Figs 3F and S5). The latter is especially true for mutation to bulkier residues in crowded environments such as T382W (Figs 3F and S4). In contrast, the choice of *nloops* of 1, 2 and 3 has no significant impact on the ΔΔG results in all cases examined (S5 Fig), indicating that the default of 1 iteration of rotamer searching is sufficient. Based on the above analysis, a repacking radius of 12 Å and a *nloop* of 1 were used to calculate Rosetta ΔΔG values for all 20 canonical amino acid substitutions (including the wild-type) at all positions of all five proteins.

## Effects of biophysics-based features in VEP performance

Armed with the Rosetta energy terms and MD-derived RMSF profiles, we examined the effects of including these biophysics-based features on VEP performance. Models based on LR, NN, CNN, and GCN were trained on the five DMS datasets individually with or without adding the biophysics-based features. All the biophysics-based features are added to the input layer in the with-biophysics models, while the models without biophysics only contain the AAIndex principal components and one-hot encoding for residues in the protein sequence (**Fig 2**). The performance was evaluated on three tasks including the overall prediction (on randomly selected testing datasets), mutational extrapolation as well as the positional extrapolation, and single mutation variants to higher-order variant extrapolation (see Methods). Five independent repeats for each splitting scheme were performed.

With random splitting, the training and testing splits have similar coverage of mutation types or sites, particularly due to the presence of many higher-order variants in the DMS datasets (S1A Fig). For the GB1 and Pab1 datasets with high mutation coverages (98% and 85%, respectively), all four ML models, with or without adding biophysics-based features, achieved nearly perfect Pearson's correlation coefficient ($r \sim 0.92 - 0.99$) on both training and testing datasets on all five individual splits (**Figs 4** and S7 **and** Table 2). However, for the Ube4b and Bgl3 datasets with only 63% and 38% mutation coverages (S1B Fig), all models were only able to achieve very modest $r \sim 0.59 - 0.65$ on training and $r \sim 0.51 - 0.56$ correlations on testing, with only marginal improvements when including biophysics-based features. This modest performance is somewhat surprising for Ube4b given its good 63% mutation coverage. It may be attributed to its poorly packed

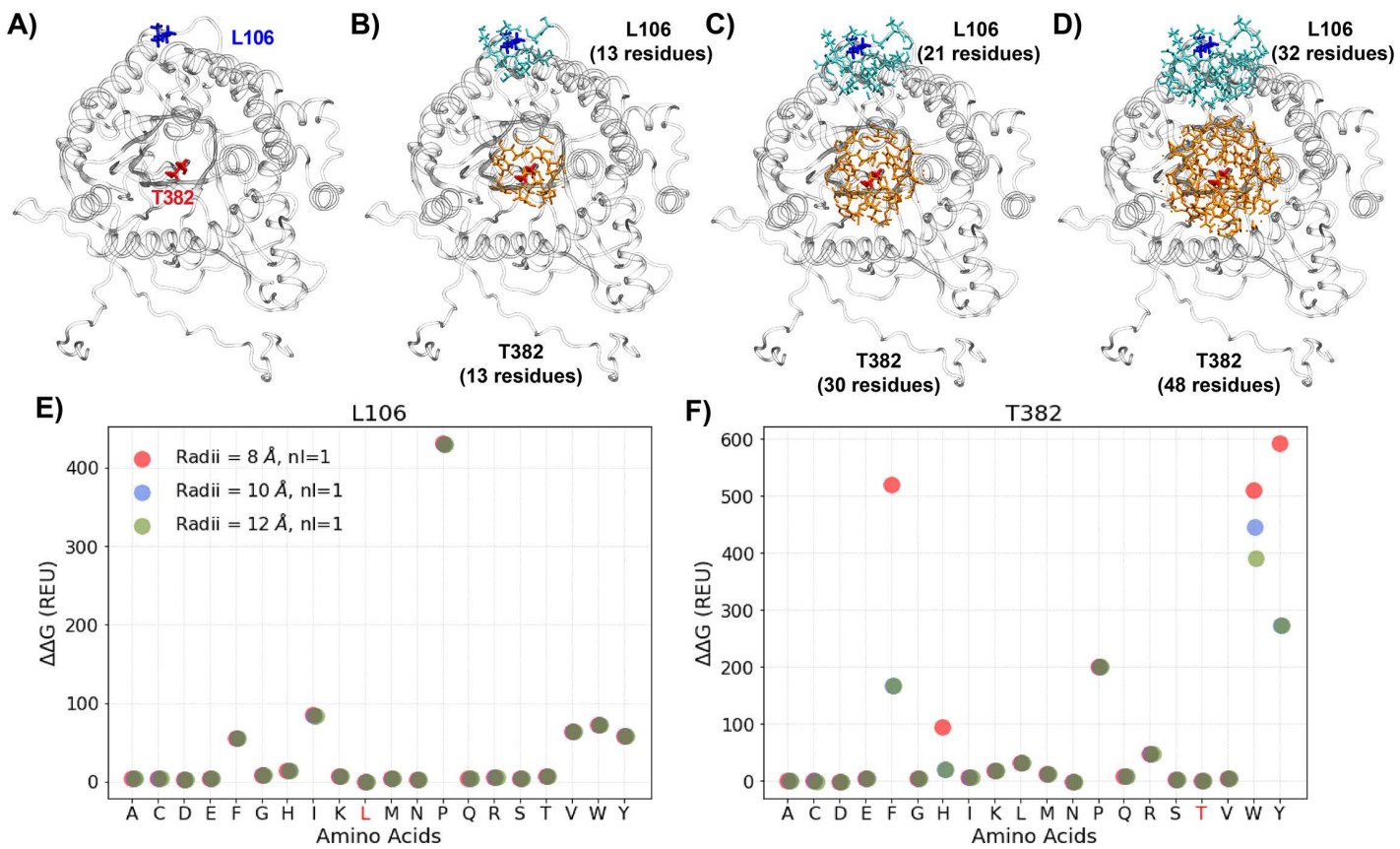

**Fig 3. Rosetta local optimization and energy evaluation of site-specific mutations using PackRotamerMover. A)** The structure of protein Bgl3 in grey cartoon, with a solvent-exposed residue L106 and a buried residue T382 shown in blue and red sticks, respectively. **B-D)** Sidechains within 8 Å **(B)**, 10 Å **(C)**, and 12 Å **(D)** of the two mutation sites, L106 and T382, shown in blue and orange sticks, respectively. The numbers of residues selected by different repacking radii are noted. **E, F)** The ΔΔG values (in REU) for the 20 amino acid substitution at site L106 and T382, calculated using three repacking radii and $nloop = 1$. The wild-type residues are marked in red in the x-axis.

structure with long loops (~30% of the sequence) (**Fig 1C**). As a result, Ube4b is highly dynamic (see RMSF in **S2C Fig**) particularly the N-terminal region (residue 1–32), rendering it difficult to accurately capture mutational impacts on structure and protein energetics. To further examine whether the low correlation is primarily driven by highly dynamic loops, we split the mutation sites from the test set into structured and loop regions based on secondary structure calculations, as shown in S8 Fig. Under the random split scheme, both models with and without biophysics-based features exhibit a clear performance difference between variants in structured and loop regions. Across all models, variants within the structured regions show higher correlation than those in the loop regions. These results support the notion that a primary limitation of ML models for protein Ube4b arises from large-scale conformational dynamics.

Interestingly, even though the avGFP dataset has only 43% mutation coverage, all models trained on either with or without biophysics-based features achieved generally good correlation ($r \sim 0.94 - 0.97$) with the LR model showing a slightly lower correlation ($r \sim 0.82$) (Figs 4 and S7). This may be attributed to the tightly packed beta-barrel structure (**Fig 1D**) and the elevated presence of high-order variants (S1A Fig). Further analysis of the mean absolute errors (MAEs), however, shows that the addition of biophysics-based terms reduces the MAE for LR and NN models for single mutation variants in protein GB1 and Pab1 datasets (S13 Fig). Nonetheless, LR and NN frequently underperform compared to CNN and GCN, especially for variants with large numbers of mutations (e.g., S13D Fig). With all five cases considered,

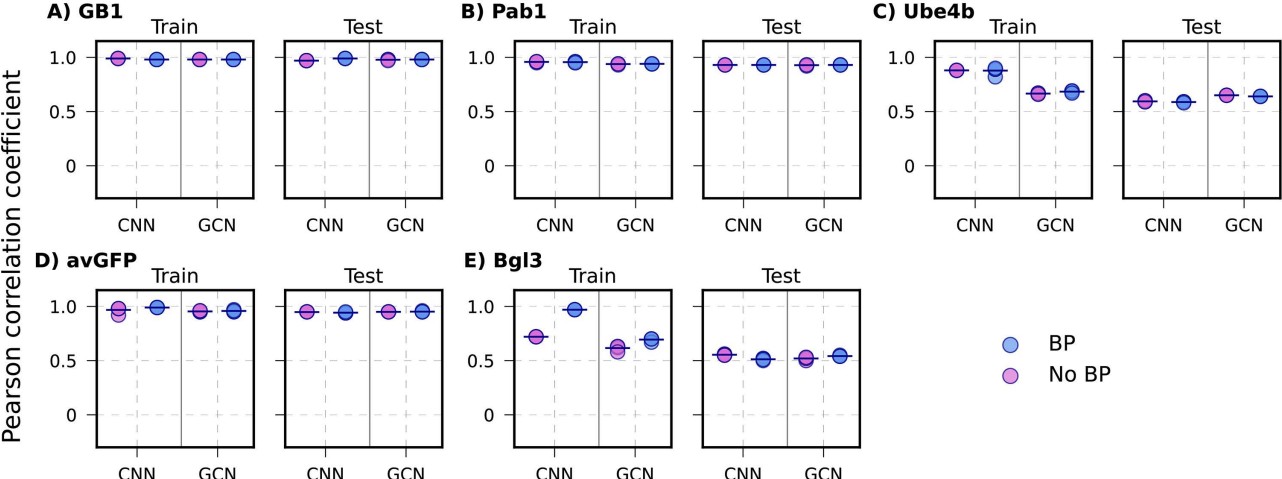

**Fig 4. Performance of CNN and GCN models on random split.** The 5 proteins are arranged based on decreasing mutational coverage. **A)** GB1, **B)** Pab1, **C)** Ube4b, D) avGFP, and **E)** Bgl3 for with biophysics-based models (BP) and without-biophysics models (No BP). Each point indicates one of the 5 random train/test, with the mean shown as black lines.

**Table 2. Average Pearson correlation coefficients for random, mutational, and positional splits across the four ML models, evaluated without biophysics features (No BP), with biophysics-based features (BP), and with BP as well as LLR (BP+LLR).**

|  |  | Random | | | | Mutational | | | | Positional | | | |
|---|---|---|---|---|---|---|---|---|---|---|---|---|---|
|  |  | LR | NN | CNN | GCN | LR | NN | CNN | GCN | LR | NN | CNN | GCN |
| GB1 | No BP | 0.92 | 0.97 | 0.97 | 0.97 | 0.68 | 0.80 | 0.89 | 0.90 | -0.01 | -0.02 | 0.05 | 0.37 |
|  | BP | 0.92 | 0.98 | 0.99 | 0.97 | 0.80 | 0.85 | 0.89 | 0.90 | -0.17 | -0.07 | **0.43** | 0.40 |
|  | BP+LLR | – | – | – | – | 0.81 | 0.85 | 0.90 | 0.91 | 0.15 | 0.12 | **0.42** | 0.65 |
| Pab1 | No BP | 0.90 | 0.92 | 0.93 | 0.93 | 0.44 | 0.67 | 0.76 | 0.71 | 0.01 | 0.08 | 0.17 | 0.03 |
|  | BP | 0.91 | 0.94 | 0.93 | 0.93 | 0.69 | 0.70 | 0.76 | 0.74 | 0.10 | 0.08 | **0.48** | **0.57** |
|  | BP+LLR | – | – | – | – | **0.82** | 0.84 | 0.85 | 0.84 | 0.25 | **0.58** | **0.69** | **0.71** |
| Ube4b | No BP | 0.60 | 0.65 | 0.59 | 0.65 | 0.19 | 0.31 | 0.45 | 0.37 | -0.03 | 0.02 | 0.08 | 0.10 |
|  | BP | 0.60 | 0.65 | 0.59 | 0.64 | **0.40** | 0.43 | 0.46 | 0.44 | -0.00 | -0.04 | 0.21 | 0.25 |
|  | BP+LLR | – | – | – | – | **0.47** | 0.49 | 0.50 | 0.49 | 0.04 | 0.15 | **0.40** | 0.37 |
| avGFP | No BP | 0.82 | 0.96 | 0.94 | 0.95 | 0.06 | 0.28 | 0.64 | 0.52 | -0.01 | -0.06 | 0.15 | 0.07 |
|  | BP | 0.82 | 0.97 | 0.94 | 0.95 | **0.61** | **0.64** | 0.79 | 0.74 | 0.08 | 0.28 | **0.69** | **0.63** |
|  | BP+LLR | – | – | – | – | **0.60** | **0.63** | 0.77 | 0.76 | 0.03 | 0.04 | **0.66** | **0.59** |
| Bgl3 | No BP | 0.52 | 0.49 | 0.55 | 0.52 | 0.02 | 0.10 | 0.18 | 0.30 | 0.00 | 0.01 | 0.11 | 0.10 |
|  | BP | 0.56 | 0.56 | 0.51 | 0.54 | **0.33** | 0.33 | 0.33 | 0.34 | 0.00 | -0.02 | 0.31 | 0.22 |
|  | BP+LLR | – | – | – | – | **0.34** | 0.34 | 0.40 | 0.37 | 0.01 | 0.08 | **0.42** | 0.37 |

including biophysics-based features only demonstrated marginal improvement on the training and testing performance for all models with random splits.

## Biophysics-based features improve VEP performance against small datasets

To further evaluate model robustness against the training set size, we trained series of CNN and GCN models with and without biophysics-based features for all proteins using only 100 randomly selected datapoints up to 0.9 fraction of the total datasets. The results, summarized in S9 Fig, clearly show that models trained with biophysics-based features

generally perform better than the without-biophysics counterparts when only seeing the limited training dataset size, especially when the sizes are less than 1000 (with low to moderate *r*). It is interesting to note that, in the case of Ube4b, CNN outperformed GCN. This suggests that if the protein structure contains many flexible loops, including structural information provides little to no advantage to the model performance, especially when trained with a limited number of data points.

Besides the DMS dataset size and mutation coverage, the quality of fitness or functional scores can also impact VEP training and performance. The fitness scores were mainly calculated using read frequencies from sequencing of the initial library and the isolated variants after the functional selection. There is a tradeoff between the dataset size and average reads per variant when the budget for the total sequencing number is fixed. We designed resampling experiments of the most complete DMS dataset of GB1 following the same protocol outlined in the NN4DMS study [41] and evaluated the performance of CNN and GCN models trained with biophysics-based features. The resample experiment generated DMS subsets with varying library sizes and numbers of reads from the original dataset [52,54]. As summarized in Fig 5, both small library sizes (thus not enough mutation coverage) and small numbers of reads (thus unreliable fitness scores) can dramatically diminish the accuracy of trained VEPs. However, including biophysics-based features clearly allow both CNN and GCN models to perform more robustly as the quality of the DMS dataset deteriorates due to either small library size or DNA reads per variant (lower left corners in Fig 5). Compared to the best CNN models previous reported in the NN4DMS study [41] and the biophysics-based CNN, when number of sequencing reads are low almost every protein library size has been increased modestly for biophysics-based CNN. For other larger sequencing reads performance of both the models are similar overall. Similarly in the case of the biophysics-based GCN, showed moderate performance improvements for low sequencing read counts and across protein libraries; however, with protein library size 5e4 and 1e5, performance decreased slightly. Curiously, both CNN and GCN architecture appear to perform similar overall. Taken together, for random splits where the training and testing data have high overlaps, including biophysics-based features marginally affects VEP performance with large DMS datasets, but can substantially improve the robustness of trained models when the mutational coverage is limited and/or the number of DNA sequencing reads is small.

## Biophysics significantly enhances mutational extrapolation

A major challenge identified from the previous NN4DMS study [41] is to reliably predict the effects of novel mutation types or mutation positions not seen in training. In the mutational extrapolation task, a VEP model trained on several mutation types on specific sites is asked to make predictions for other mutation types on those sites (see Methods). As summarized in **Figs 6** and **S10** and **Table 2**, all models are still performing well with the GB1 dataset due to its 98% sequence percentage converge. For other protein datasets, adding biophysics-based features in general improved the performance of all four ML models on their testing sets, especially for avGFP. For models with lower complexity such as LR and NN, biophysics-based features can dramatically improve prediction, such as LR models for the Pab1 (*r*: from 0.44 to 0.69), Ube4b (*r*: from 0.19 to 0.40), avGFP (*r*: from 0.06 to 0.61) and Bgl3 (*r*: from 0.02 to 0.33) datasets and NN models in the avGFP (*r*: from 0.28 to 0.64) and Bgl3 (*r*: from 0.1 to 0.33) datasets (S10 Fig). For models with higher complexities such as CNN and GCN, those biophysics-based features still could help achieve slightly better performances, such as in avGFP (from 0.52 to 0.74), and Bgl3 (from 0.18 to 0.33) datasets (**Fig 6**). These results indicate that biophysics-based features provide important and useful information for mutational extrapolation task.

To further examine if including sequence evolution information could further improve model performance, we incorporated LLR scores from four ESM-2 models of increasingly large sizes (***see Methods*** for details) and found that the 33M-parameter version provided the best overall performance. Therefore, we selected ESM-2 (33M) as the final model for estimating LLR scores. Notably, integrating evolutionary features led to small but consistent improvement in predictive performance across all proteins compared to the biophysics-only model (**Table 2** and Figs 6-7).

We also conducted ablation studies using protein Pab1 to understand the contribution of different biophysics-based features towards mutational extrapolation performance (see Methods). As summarized in S11 Fig, all models without Rosetta-based energetic features perform slightly worse than the other variants. Performances remain reasonably strong across all the models

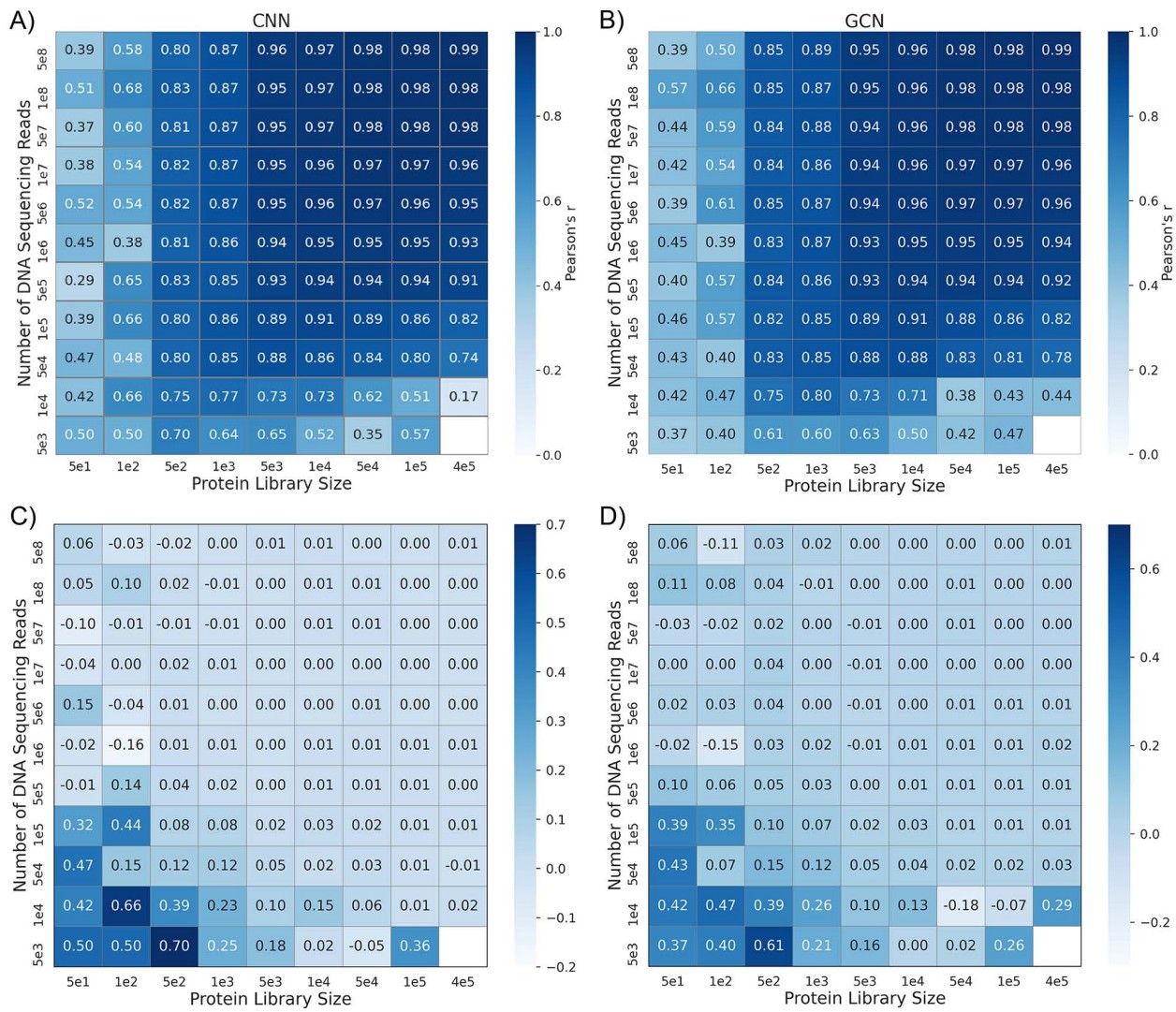

**Fig 5. Dependence of VEP performance on the library size and number of sequencing reads.** A, **B)** Pearson's correlations for CNN and GCN models trained with biophysics-based features using different combination of the library size and number of sequencing reads for the protein GB1 DMS dataset. The empty box represents the combination where the number of variants was not enough for the experiment. C, **D)** Differences in Pearson's correlation coefficients between the above CNN and GCN models and previously CNN models trained without biophysics-based features [41].

including without biophysics-based features. Interestingly, removing RMSF feature appears to slightly improve the performance than the full NN, CNN and GCN models trained with all biophysics-based features. The LR model was the only architecture for which incorporating biophysics-based features led to a substantial improvement compared to without biophysics model for protein Pab1. In this case, all the ablation variant models with biophysics-based features have resulted in improved performance, suggesting that RMSF features alone can also mitigate the mutational extrapolation challenge. Overall, the ablation analysis suggests that all biophysics-based features are of high quality and contribute to improved extrapolation.

## Biophysics significantly enhances positional extrapolation

Positional extrapolation is the most challenging task since it requires a VEP model trained only on certain specific mutation sites to make correct predictions on mutations to sites it has never seen during training. All four ML models, when

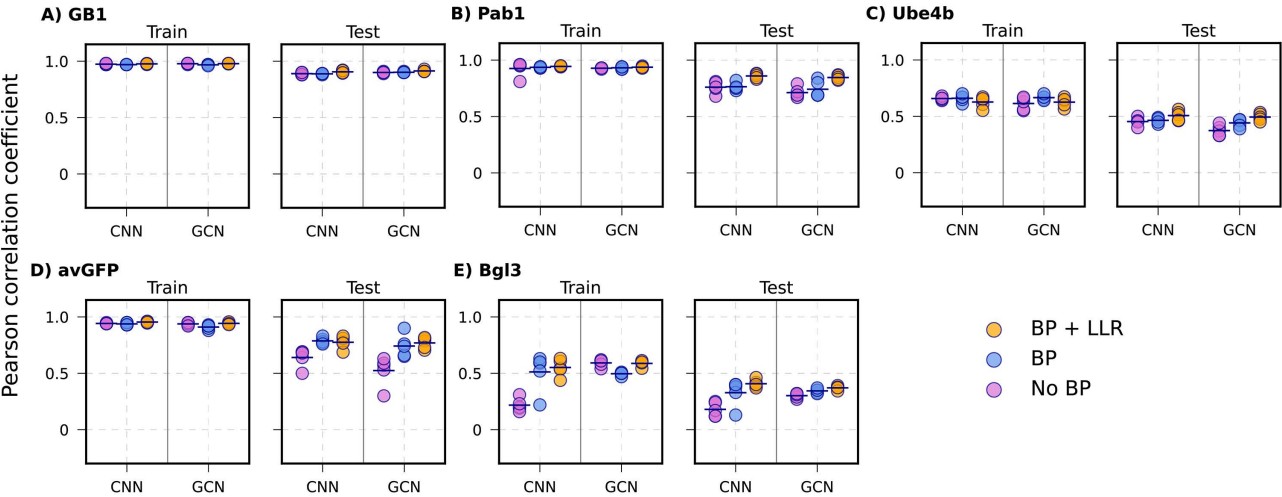

**Fig 6. Performance of CNN and GCN models trained without biophysics (No BP), with biophysics (BP), and with biophysics as well as LLR (BP+LLR) for mutational extrapolation.** The 5 proteins are arranged with decreasing mutational coverage. **A)** GB1, **B)** Pab1, **C)** Ube4b, D) avGFP, and **E)** Bgl3. Each point indicates one of the 5 random mutational train/test splits, with the mean shown as black lines.

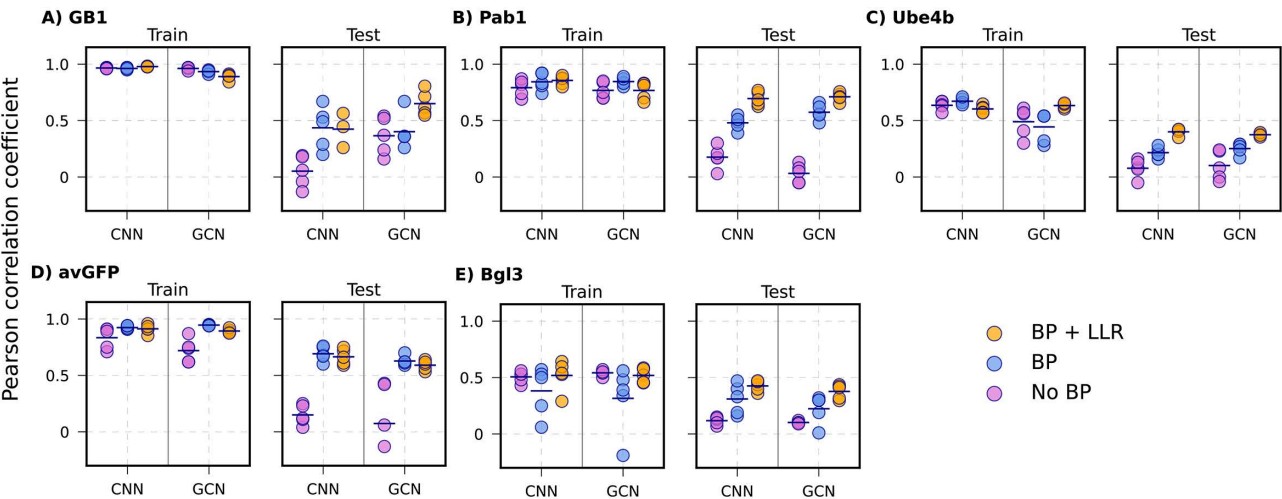

**Fig 7. Performance of CNN and GCN models trained without biophysics (No BP), with biophysics (BP), and with biophysics as well as LLR (BP+LLR) for positional extrapolation.** The 5 proteins are arranged with decreasing mutational coverage. **A)** GB1, **B)** Pab1, **C)** Ube4b, D) avGFP, and **E)** Bgl3. Each point indicates one of the 5 random positional train/test splits, with the mean shown as black lines.

trained without biophysics-based features, show either no correlation (r~0) or very low correlation (r<0.2) on the testing set, except for GCN in the case of GB1 (Figs 7 and S12), highlighting the difficulty of this task and also the merit of incorporating protein structure information. Strikingly, incorporating biophysics-based features significantly enhances the ability of all VEP models to handle positional extrapolation, especially with CNN and GCN (Fig 7). For example, in the case of GB1 CNN (from 0.05 to 0.43), Pab1 (0.17 to 0.48), avGFP (0.15 to 0.69) dataset, and GCN in the Pab1 (0.03 to 0.57) and avGFP (0.07 to 0.63) datasets. In other cases where the protein itself has more complexity to model, adding biophysics-based features can still marginally improve their performance, such as CNN and GCN in the Ube4b (from

0.08-0.10 to 0.21-0.25) and Bgl3 (from 0.1 to 0.22-0.31) dataset. This strongly supports that biophysics-based features are indeed beneficial for positional extrapolation, especially for well-structured proteins with low sequence percentage coverages (such as avGFP). The MAE comparison also shows that incorporating biophysics-based terms reduces the MAE across all models and for all types of variants as shown in S13, S14, and S15 Figs. Notably, including LLR derived from ESM-2 (33M) led to significant improvements in performance across all proteins compared to the biophysics-only models, with the exception of protein avGFP. These results indicate that evolutionary information can meaningfully enhance predictive performance for positional extrapolation.

Similar ablation studies were conducted to analyze the contributions of various biophysics features to positional extrapolation using protein Pab1. We focus on CNN and GCN models, as both LR and NN model largely fail for positional extrapolation (S12 Fig). As summarized in S16 Fig, all GCN models incorporating biophysics-based features exhibit very similar performance, whereas all CNN model variants using subsets of the biophysics-based features show substantially lower performance compared with the full model with the all biophysics-based features. The implication is that biophysics-based features act synergistically to enhance the ability of positional extrapolation.

## Performance on single-mutation to high-order variant extrapolation

Another extrapolation splitting scheme examined is where models are trained on single mutation variants and then evaluated on the higher-order variants (**see Methods**). Under this splitting scheme, all ML exhibit very similar performance with and without biophysics-based features across all the proteins (Figs 8 and S17). For GB1 and Pab1, all the models achieve comparable performance, with mean $r \sim 0.88$ and $\sim 0.80$, respectively. Similarly, for Ube4b and the Bgl3, all models show similar performance (mean $r \sim 0.40$ for both proteins). However, for avGFP, models incorporating biophysics-based features perform better than those without, especially for CNN and GCN architectures (without biophysics $r \sim 0.57$ and with biophysics $r \sim 0.70$). These results indicate that cooperativity is rarely significant in higher-order variants. That is, the effects of individual mutations are mostly additive. Similar observations have been made in previous studies [48]. The lack of cooperativity also supports our model design that requires only Rosetta energetics of all possible single mutations (**see** Methods). We note that the performance of single-mutation to high-order variant extrapolation of our current models with

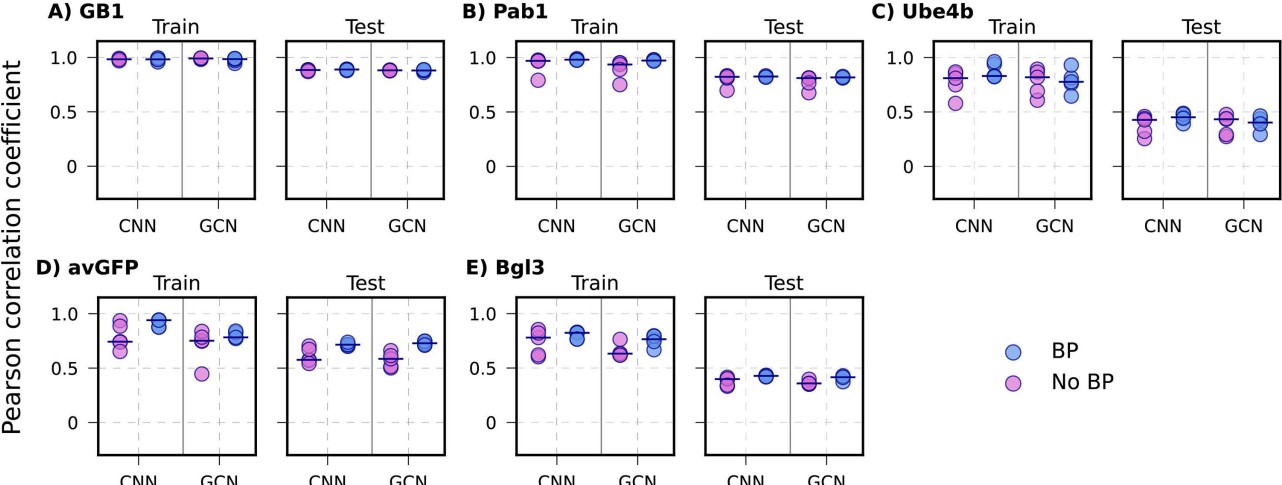

**Fig 8. Performance of CNN and GCN models with and without BP in the single mutation variant to higher-order variant extrapolation.** The 5 proteins are arranged from the largest to smallest sequence space coverage of full dataset. **A)** GB1, **B)** Ube4b, C) avGFP, **D)** Pab1, and **E)** Bgl3. Each point indicates one of the 5 random train/test splits, and the mean of the 5 replicates are marked using black lines.

biophysics is comparable to the transformer-based METL-local model [48] except for avGFP. For avGFP, METL-local achieves a higher Spearman correlation coefficient of ~ 0.80 [48], compared to that of ~ 0.73 at best with GCN with biophysics in this work.

## Discussion

Characterization of protein sequence-to-function relationships is critical to study protein evolution, understand genetic diseases, and design proteins with enhanced or new functions. However, the inherent complexity from large sequence space, structural variability, and conformational dynamics makes this mapping task extremely challenging. The availability of sequencing data, protein structures, and high-throughput functional assays has allowed the training of machine learning models to infer protein sequence-function relationships and predict variation effects. In this work, we develop and evaluate a simple and efficient approach for incorporating biophysics-based features to train VEP models with superior robustness against dataset limitations and ability to predict effects of novel mutational types or positions. This approach only requires the number of residues x 19 fast Rosetta structural relaxation and energy evaluations, each of which takes only seconds for a typical protein. The effectiveness of the biophysics-inspired VEP development approach has been evaluated using existing DMS datasets of five proteins with diverse size, topology, and mutation coverage. The results strongly support the effectiveness of incorporating biophysics to improve VEP performance and overcome DMS dataset limitations, especially for positional extrapolation with both CNN and GCN.

Our results provide several key insights for building robust VEP models trained on DMS datasets. Firstly, the sequence percentage coverage is critical. For the GB1 dataset, which has 98% coverage, all models, including simple LR performed well in the random and mutational splitting schemes. Whereas for the Bgl3 datasets, which has only 38% coverage, all VEP models showed medium to low Pearson correlation coefficient, with an average of ~0.54 for random and ~0.33 for mutational. Arguably, such data scarcity could only be effectively mitigated in general by acquiring more functionally labelled data points. Secondly, incorporating the physical principles of protein structure and interaction is an effective strategy for overcoming data scarcity problem in VEP development. The physical principles could be captured by estimating mutational impacts on the protein energetics and by considering the structural flexibility as derived from short MD simulations. Models trained with these biophysics-based features can dramatically enhance mutational and positional extrapolations for well-structured proteins, such as for the avGFP dataset with only 43% mutation coverage. Last but not the least, the structure and dynamics of the protein are also important factors that affect VEP performance. For example, existing variants may not effectively inform on new ones for highly dynamic proteins, such as in the case of Ube4b. In these cases, obtaining reliable biophysics-based features is much more challenging. The current approach focuses on capturing localized structural and energetic impacts of mutations changes and thus could only help marginally. Exploring the use of advanced machine learning techniques, such as transformer-based models, in combination with biophysics-based features, could further improve extrapolation capabilities.

Even though protein stability is an important factor for functional fitness, it is not always the sole or even a major determinant. Mutations can affect protein function through other mechanisms, such as disruption of binding or active sites, changing the expression level, and affecting protein dynamics. As such, biophysics-based features may be more informative for certain types of fitness measurements than others [76]. Along this line, evolutionary information such as those captured in LLMs can provide complementary information to maximize VEP performance. Indeed, we observe that incorporating LLR scores from the ESM model improves performance for most of the proteins studied, supporting the synergetic enhancement of combining biophysics-based features with evolutionary information. Interestingly, despite the simplicity of our strategy of directly including biophysical properties and LLR scores as the input features (Fig 2), both CNN and GCN models perform at levels comparable to more advanced transformer-based models such as METL-local [48] and GVP-MSA [32] for both mutational and positional extrapolations (S18 Fig), despite significant variability in performance for different proteins in the more challenge case of positional extrapolation.

 

In conclusion, our study demonstrates that incorporating biophysics-based features into deep learning models significantly enhances their ability to predict protein function from sequence data, especially in extrapolation scenarios. This approach holds great promise for advancing protein engineering and understanding protein-related diseases as well as membrane proteins and transmembrane channels. This is also highlighted in our recent work where biophysics-based features allow a reliable predictor of activation voltage of BK channels to be derived using less than 500 single-mutation variants [50]. Note that an important advantage if physics-informed ML approaches is that they can be readily extended to account for post-translational modifications, non-canonical amino acids, and other complex biochemical effects, where sequence centric approaches such as GVP-MSA will not be applicable. More reliable VEPs will provide valuable tools for researchers, aiding the prioritization and comprehensive understanding of the genetic variations for various diseases and protein engineering. They can also contribute to identifying potential drug targets and understanding them under various conditions.

## Supporting information

**S1 Fig. Distribution and coverage of variants in DMS datasets.** A) Distribution of various containing different number of mutations in each of the five datasets. B) Number of unique mutations sampled for each protein. The coverage percentage, listed under the x-axis, is defined as the number of unique mutations (+ wildtype) divided by the total possible number (20 x sequence length). The sequence length of each protein is given in parathesis next to the protein name.
(TIF)

**S3 Fig. Optimized local side chain configurations upon mutating solvent exposed L106 of protein Bgl3 to all 19 other amino acids, generated using the PackRotamerMover protocol with a repacking radius of 12 Å and nloops of 1.** The protein backbone is shown in white cartoon and all residues within the repacking radius are represented as yellow sticks with residue L106 highlighted in red. Elements are colored as follows: C (yellow), N (blue), O (red), and S (orange).
(TIF)

**S4 Fig. Optimized local side chain configurations upon mutating buried T382 of protein Bgl3 to all 19 other amino acids, generated using the PackRotamerMover protocol with a repacking radius of 12 Å and nloops of 1.** The protein backbone is shown in white cartoon and all residues within the repacking radius are represented as cyan sticks with T382 highlighted in red. Elements are colored as follows: C (cyan), N (blue), O (red), and S (yellow).
(TIF)

**S5 Fig. Convergence of ΔΔG with different repacking radii and nloop values in PackRotamerMover.** The ΔΔG (in REU) values for the 20 amino acid substitution was calculated for solvent exposed L106 (A) and buried T382 (B).
(TIF)

**S6 Fig. Distribution of all the 19 Rosetta terms and ΔΔG energies.** The blue bar plots indicate term used as the input features and red bar plots indicate terms discarded as the input features.
(TIF)

**S7 Fig. Performance of LR and NN models with or without biophysics in the random splitting scheme.** The 5 proteins are arranged from the largest to smallest sequence space coverage: A) GB1, B) Pab1, C) Ube4b, D) avGFP, and E) Bgl3 for with biophysics-based mode (BP) and without-biophysics model (No BP). Each point indicates one of the 5 random train/test splits, and the mean of the 5 replicates are marked using black lines.
(TIF)

**S8 Fig. Pearson correlation coefficient of VEP models for the Ube4b dataset, divided into the structured and loop regions.** A) Results obtained with BP model and B) Results obtained without BP model for 3 splitting schemes.
(TIF)

**S9 Fig. Performance of CNN and GCN models with and without biophysical features with respect to training random split fraction.** The training dataset was increased from 100 datapoints to 0.9 fraction of the total dataset of each protein. The solid line indicates CNN, and dashed line indicates GCN model.
(TIF)

**S10 Fig. Performance of LR and NN models without biophysics (No BP), with (BP), and biophysics with LLR (BP+LLR) in the mutational splitting scheme.** The 5 proteins are arranged from the largest to smallest sequence space coverage: A) GB1, B) Pab1, C) Ube4b, D) avGFP, and E) Bgl3. Each point indicates one of the 5 random train/test splits, and the mean of the 5 replicates are marked using black lines.
(TIF)

**S11 Fig. Ablation study of the biophysics-based features for mutational splitting scheme on the Pab1 protein.** The ablation study is done for the four different models A) GCN, B) CNN, C) NN, and D) LR. Each point represents one of five random train/test splits, and the mean across the five replicates is indicated by black lines. Models were trained using different feature subsets: no biophysics (No BP), no RMSF, no Rosetta, no Lennard–Jones and no ΔΔG (No LJ), no electrostatics and no ΔΔG (No Elec), no solvation and no ΔΔG (No Solv), and all biophysical features included (All).
(TIF)

**S12 Fig. Performance of LR and NN models without biophysics (No BP), with (BP), and biophysics with LLR (BP+LLR) in the positional splitting scheme.** The 5 proteins are arranged from the largest to smallest sequence space coverage. A) GB1, B) Ube4b, C) avGFP, D) Pab1, and E) Bgl3. Each point indicates one of the 5 random train/test splits, and the mean of the 5 replicates are marked using black lines.
(TIF)

**S13 Fig. Mean absolute error with respect to number of mutations in variant for models trained with and without biophysics using random splitting.** Each color point represents one of the 5 random train/test splits, for LR (salmon), NN (orchid), CNN (turquoise) and GCN (green) models.
(TIF)

**S14 Fig. Mean absolute error with respect to number of mutations in variant for models trained with and without biophysics for mutational extrapolation.** Each color point represents one of the 5 random splits for LR (salmon), NN (orchid), CNN (turquoise) and GCN (green) model.
(TIF)

**S15 Fig. Mean absolute error with respect to number of mutations in variant for models trained with and without biophysics using random splitting for positional extrapolation.** Each color point represents one of the 5 random splits for LR (salmon), NN (orchid), CNN (turquoise) and GCN (green) model.
(TIF)

**S16 Fig. Ablation study of the biophysics-based features for positional splitting scheme on the Pab1 protein.** The ablation study is done for the four different models A) GCN, B) CNN, C) NN, and D) LR. Each point represents one of five random train/test splits, and the mean across the five replicates is indicated by black lines. Models were trained using different feature subsets: no biophysics (No BP), no RMSF, no Rosetta, no Lennard–Jones and no ΔΔG (No LJ), no electrostatics and no ΔΔG (No Elec), no solvation and no ΔΔG (No Solv), and all biophysical features included (All).
(TIF)

**S18 Fig. Performance of CNN and GCN models with biophysics (BP) and additional LLR augment (BP+LLR) against METL-local [1] and GVP-MSA (multi-protein) [2] on mutational (A) and positional (B) extrapolation tests.**

Performance is evaluated using the average Spearman correlation coefficient. Results for METL-local and GVP-MSA (multi-protein) are directly taken from their original publications. For GVP-MSA, only single-mutation variants were considered.
(TIF)

**S1 Table. Possible input features.** The features are divided into four categories: (1) One-hot encoding of 21 variables representing 20 amino acids and 1 stop codon; (2) AAIndex features consisting of 19 numerical values based on physicochemical properties [3]; (3) Rosetta energetics with 20 energy terms related to protein stability and folding, including attractive forces (fa_atr), van der Waals repulsive forces (fa_rep, fa_intra_rep), solvation energy (fa_sol, Fa_intra_sol_ xover4, lk_ball_wtd), dielectric electrostatics (fa_elec), Proline ring closing energy (pro_close), disulfide statistical energies (dslf_fa13), and hydrogen bonding contributions (hbond_sr_bb, hbond_lr_bb, hbond_bb_sc, hbond_sc); and (4) the Root mean square fluctuation (RMSF) derived from atomistic simulations, capturing protein flexibility and represented by a single feature. Note that only 8 of the 20 Rosetta energy terms are selected to be used in training final VEPs reported in this work (highlighted in italic).
(XLSX)

**S2 Table. Hyperparameters used for training various models on five different protein datasets.** The table lists the batch size (32, 64, 128, 1024, 2048), learning rate (0.005, $1 \times 10^{-3}$, $1 \times 10^{-4}$), number of dense layers (1, 2, 3), and number of nodes for NN models (100, 200). For CNN models, the number of convolutional layers and filter sizes are reported, while for GCN models, the graph thresholds (in Å) are specified. If mutational or positional splits used different hyperparameters, they are indicated separately within the corresponding cell.
(XLSX)

**S3 Table. Average Pearson correlation coefficients for single mutation variants to higher-order variants splits across the four ML models with and without biophysics-based features (BP).**
(XLSX)

## Acknowledgments

The authors thank Sam Gelman and Anthony Gitter for helpful discussions.

## Author contributions

**Conceptualization:** Shrishti Barethiya, Hui Guan, Jianhan Chen.

**Data curation:** Shrishti Barethiya.

**Funding acquisition:** Jianhan Chen.

**Investigation:** Shrishti Barethiya, Jian Huang, Clarice Stumpf.

**Methodology:** Shrishti Barethiya, Jian Huang, Xiao Liu, Hui Guan, Jianhan Chen.

**Resources:** Jianhan Chen.

**Supervision:** Jianhan Chen.

**Validation:** Shrishti Barethiya, Clarice Stumpf.

**Visualization:** Shrishti Barethiya.

**Writing – original draft:** Shrishti Barethiya, Jian Huang.

**Writing – review & editing:** Shrishti Barethiya, Jian Huang, Clarice Stumpf, Hui Guan, Jianhan Chen.

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
