## [Decision Letter · Decision Letter 0]

11 Dec 2025

PCOMPBIOL-D-25-02332

Overcoming Extrapolation Challenges of Deep Learning by Incorporating Physics in Protein Sequence-Function Modeling

PLOS Computational Biology

Dear Dr. Chen,

Thank you for submitting your manuscript to PLOS Computational Biology. After careful consideration, we feel that it has merit but does not fully meet PLOS Computational Biology's publication criteria as it currently stands. Therefore, we invite you to submit a revised version of the manuscript that addresses the points raised during the review process.

We look forward to receiving your revised manuscript.

Kind regards,

Mingyue Zheng

Guest Editor

PLOS Computational Biology

Nir Ben-Tal

Section Editor

PLOS Computational Biology

**Journal Requirements:**

At this stage, the following Authors/Authors require contributions: Shrishti Barethiya, Jian Huang, Xiao Liu, Hui Guan, and Jianhan Chen. Please ensure that the full contributions of each author are acknowledged in the "Add/Edit/Remove Authors" section of our submission form.

5) We have noticed that you have uploaded Supporting Information files, but you have not included a list of legends. Please add a full list of legends for your Supporting Information files after the references list.

**Reviewers' comments:**

Reviewer's Responses to Questions

**Comments to the Authors:**

Reviewer #1: This manuscript presents a valuable approach to overcoming the extrapolation limitations of deep learning in protein sequence-function modeling by incorporating biophysics-based features (Rosetta energy terms and MD-derived RMSF). The topic is highly relevant to protein engineering and variant effect prediction, and the experimental design using five diverse proteins with varying DMS coverage provides a solid foundation for evaluation. However, several critical issues and gaps need to be addressed to enhance the rigor, generalizability, and depth of the study, as outlined below.

1. The current work relies heavily on high-quality experimental PDB structures as initial inputs for simulations. To fully assess the method’s applicability in real-world scenarios where experimental structures are scarce, the authors should test the sensitivity of the approach to initial structure quality. Specifically, evaluating performance using computationally predicted structures (e.g., AlphaFold2 models, especially those with moderate confidence scores) would help clarify whether the biophysics-integrated strategy is robust to structural inaccuracies and expand the method’s practical utility.

2. While mutational and positional extrapolation are well evaluated, the manuscript lacks an assessment of the model’s ability to extrapolate from single variants to higher-order (e.g., double, triple) variant effects. This is a critical use case for VEP tools in protein engineering. The authors should supplement this analysis, including a direct comparison with simple additive models, to demonstrate whether the biophysics-based features capture non-trivial epistatic interactions that improve higher-order variant prediction.

3. The discussion of biophysics-based feature (BP) performance across proteins with different DMS coverage does not account for confounding factors, particularly the inherent variability in the "fitness" metrics measured by DMS across different protein systems. For example, fitness could reflect molecular binding, catalytic activity, or stability—processes that vary in their sequence-function mapping complexity. Failing to disentangle the impact of coverage from that of fitness metric heterogeneity makes it difficult to draw definitive conclusions about BP’s effectiveness across diverse biological contexts. The authors should address this by, for instance, stratifying analysis by fitness type or including a control that normalizes for mapping complexity.

4. The study would benefit from more rigorous ablation studies to quantify the individual contributions of different BP features. Currently, it is unclear whether specific components (e.g., Rosetta energy terms vs. MD-RMSF, or individual energy sub-terms) drive the observed performance gains. Systematic removal of each feature category and comparison of model performance would clarify which biophysical information is most critical, enhancing the interpretability of the approach and guiding future method optimization.

5. For proteins with high structural flexibility (e.g., Ube4b with long loops), the improvement from BP features is modest. The authors should provide a more in-depth analysis of this limitation—e.g., whether the current BP features (focused on local interactions and static flexibility) fail to capture global conformational dynamics, and suggest potential modifications (e.g., integrating ensemble-based energy calculations or dynamic correlation features) to address this gap.

Addressing these points will strengthen the manuscript’s scientific impact and ensure that the proposed approach is thoroughly validated for broader application in protein sequence-function modeling.

Reviewer #2: In the manuscript under consideration, the authors combined biophysics-based features (Rosetta energetics and MD-derived RMSF) and deep learning architectures (CNN/GCN) to address extrapolation challenges in protein sequence-function modeling. While addressing the data scarcity is a critical goal, I have concerns regarding the novelty of the methodology, the validity of the baselines, and the biophysical assumptions employed. My specific concerns are listed as below.

1) The most critical flaw in this study is the absence of comparison with current state-of-the-art deep learning methods, specifically Protein Language Models (PLMs) (e.g., ESM-1v, ESM-2) and Inverse Folding Models (e.g., ProteinMPNN), which efficiently leverage structural information without the need for costly MD simulations. These models have demonstrated exceptional zero-shot and few-shot performance on positional extrapolation tasks without requiring explicit physical feature engineering. It is highly probable that a standard fine-tuned ESM-2 model would outperform the proposed method, rendering the complex feature engineering pipeline unnecessary.

2) The strategy of using Wild-Type (WT) RMSF profiles to represent the dynamics of mutant variants is methodologically questionable, particularly for the "extrapolation" tasks the authors aim to solve. The mutations, especially multi-point ones or those exhibiting epistasis, often significantly alter the protein's conformational ensemble and flexibility. Using WT dynamics as a static feature for all variants ignores the very physical changes that drive functional differences. Therefore, the "physics" incorporated here is an approximation that may not hold for the most challenging prediction targets (i.e., those with significant structural deviations from WT).

3) The technical contribution appears to be primarily feature engineering—simply concatenating Rosetta energy terms and RMSF values into the input layer of standard CNN/GCN architectures. In the Discussion section, the authors mentioned that combining these features as input of the Transformer-based architectures could improve performance. Given that Transformer-based architectures are now standard, sticking to older architectures (CNN/GCN) without a compelling justification limits the impact and relevance of the study.

4) While the authors claim "significant enhancement," the absolute performance in challenging cases remains poor. For example, in the Ube4b positional extrapolation task, the Pearson correlation only reaches ~0.2-0.25 even with biophysical features. This suggests that the added physical features fail to capture the necessary signal for flexible proteins. The trade-off between the high computational cost (running MD and Rosetta for feature generation) and the marginal performance gain in difficult cases is not well-justified.

Reviewer #3: This study focuses on the extrapolation challenges faced by deep learning in protein sequence-function modeling. It integrates biophysical features such as Rosetta energy terms and residue root mean square fluctuation (RMSF) derived from molecular dynamics (MD) simulations into models including convolutional neural networks (CNNs) and graph convolutional neural networks (GCNs). This integration effectively enhances the models' predictive performance for unseen mutation types (mutational extrapolation) and mutation positions (positional extrapolation), providing a highly valuable physics-informed solution to address protein variant effect prediction (VEP) problems under data scarcity scenarios. However, the following issues need to be resolved prior to publication.

1. Insufficient verification of parameter robustness for physical feature calculation. The study only optimized two parameters (repacking radius and nloop) for Rosetta’s PackRotamerMover protocol, but failed to verify the impact of key parameters such as force field version (REF2015) and side-chain library (Dunbrack) on ΔΔG calculations. Variations in parameter combinations may cause fluctuations in energy features, necessitating supplementary parameter sensitivity analysis.

2. Questionable sufficiency of MD simulation duration and conformational sampling. A 200-ns MD simulation may lead to inadequate conformational sampling for flexible proteins (e.g., Ube4b with long loops). The study did not calculate the convergence of root mean square deviation (RMSD) or adopt enhanced sampling methods (e.g., metadynamics), making it difficult to accurately characterize the intrinsic flexibility of residues. Convergence verification of simulations should be supplemented.

In addition, only a single 200-ns all-atom MD simulation was performed per protein to obtain residue Cα RMSF values, with no routine ≥3 independent replicate simulations. This risks RMSF distortion due to initial conformation dependence and random thermal motion, introducing non-specific model errors that undermine physical feature reliability and conclusion robustness. It is recommended to add at least two replicate MD simulations with identical parameters but different initial random seeds, calculate RMSF mean and standard deviation to assess feature variability, optimize simulation strategies (e.g., extended duration or enhanced sampling) if significant RMSF discrepancies exist across replicates, and retrain models using statistically averaged RMSF data to verify feature stability and improve result reproducibility and credibility.

3. Lack of rationality for the additivity assumption of multi-mutation energy effects. The model assumes that the energy effects of multi-site mutations are linearly additive across single-point mutations, yet epistasis (nonlinear interactions between mutation sites, including synergistic or antagonistic effects) is prevalent in natural proteins. The study did not evaluate the prediction errors induced by this additivity assumption, and should compare energy differences between the additive model and all-atom calculations for multi-site mutations.

4. Ambiguous graph construction logic for GCN. The GCN constructs contact maps based on Cα/Cβ atom distances, using a threshold of 6 Å for Bgl3 and 7 Å for other proteins, but does not explain the biological rationality of these thresholds. Comparisons of model performance across different distance thresholds (e.g., 5 Å, 8 Å) should be supplemented to validate graph topology optimization.

5. Only four traditional architectures (LR/NN/CNN/GCN) were employed, without integrating cutting-edge hybrid models in the AIDD field (e.g., CNN-Transformer, GVP-GNN) or attention mechanisms to capture long-range site correlations. This limits the model’s ability to address complex dependencies in high-order mutations (e.g., synergistic multi-site mutations).

6. Insufficient representativeness of model protein types. The five selected proteins exclude core CADD research targets such as membrane proteins, multi-subunit complexes, and transmembrane channels, and are all water-soluble single-domain proteins. The model’s generalizability to membrane environments and protein-ligand interaction scenarios was not verified, and the dataset should be extended to include diverse protein types.

7. Unvalidated cross-protein generalization capability. The model was trained separately for each protein (protein-specific training) and did not conduct cross-protein transfer learning experiments (e.g., training on GB1 data to predict avGFP mutations). A core value of AIDD tools lies in cross-target generalization, and cross-protein extrapolation performance data should be added.

8. Superficial mechanism analysis for poor performance on highly flexible proteins. The limited performance improvement of Ube4b was attributed to "long-loop structures," but the study did not analyze correlations between RMSF and energy features or quantify the interference of conformational dynamics on mutation effect prediction. Mechanistic research linking protein structure to model performance should be supplemented.

9. Lack of relevance to drug targets. Most selected proteins are tool proteins (e.g., avGFP, GB1), excluding disease-related targets (e.g., kinases, GPCRs). This fails to reflect the model’s application value in drug development, and case studies of at least one disease-related target protein should be added.

**Have the authors made all data and (if applicable) computational code underlying the findings in their manuscript fully available?**

The PLOS Data policy requires authors to make all data and code underlying the findings described in their manuscript fully available without restriction, with rare exception (please refer to the Data Availability Statement in the manuscript PDF file). The data and code should be provided as part of the manuscript or its supporting information, or deposited to a public repository. For example, in addition to summary statistics, the data points behind means, medians and variance measures should be available. If there are restrictions on publicly sharing data or code —e.g. participant privacy or use of data from a third party—those must be specified.requires authors to make all data and code underlying the findings described in their manuscript fully available without restriction, with rare exception (please refer to the Data Availability Statement in the manuscript PDF file). The data and code should be provided as part of the manuscript or its supporting information, or deposited to a public repository. For example, in addition to summary statistics, the data points behind means, medians and variance measures should be available. If there are restrictions on publicly sharing data or code —e.g. participant privacy or use of data from a third party—those must be specified.

Reviewer #1: Yes

Reviewer #2: Yes

Reviewer #3: None

PLOS authors have the option to publish the peer review history of their article (what does this mean? ). If published, this will include your full peer review and any attached files.). If published, this will include your full peer review and any attached files.

**Do you want your identity to be public for this peer review?** For information about this choice, including consent withdrawal, please see our For information about this choice, including consent withdrawal, please see our Privacy Policy ..

Reviewer #1: **Yes:** Mingyue ZhengMingyue Zheng

Reviewer #2: No

Reviewer #3: No

**Figure resubmission:**

**Reproducibility:**



---

## [Decision Letter · Decision Letter 1]

10 Mar 2026

Dear Dr. Chen,

We are pleased to inform you that your manuscript 'Overcoming Extrapolation Challenges of Deep Learning by Incorporating Physics in Protein Sequence-Function Modeling' has been provisionally accepted for publication in PLOS Computational Biology.

Best regards,

Mingyue Zheng

Guest Editor

PLOS Computational Biology

Nir Ben-Tal

Section Editor

PLOS Computational Biology

Reviewer's Responses to Questions

**Comments to the Authors:**

Reviewer #1: In this revision, the authors have adequately addressed the issues raised in the previous round of review. I feel that the manuscript is appropriate for publication in its current form.

Reviewer #3: The authors have generally addressed the questions and concerns raised in my review. I appreciate their careful responses and revisions.

**Have the authors made all data and (if applicable) computational code underlying the findings in their manuscript fully available?**

The PLOS Data policy requires authors to make all data and code underlying the findings described in their manuscript fully available without restriction, with rare exception (please refer to the Data Availability Statement in the manuscript PDF file). The data and code should be provided as part of the manuscript or its supporting information, or deposited to a public repository. For example, in addition to summary statistics, the data points behind means, medians and variance measures should be available. If there are restrictions on publicly sharing data or code —e.g. participant privacy or use of data from a third party—those must be specified.requires authors to make all data and code underlying the findings described in their manuscript fully available without restriction, with rare exception (please refer to the Data Availability Statement in the manuscript PDF file). The data and code should be provided as part of the manuscript or its supporting information, or deposited to a public repository. For example, in addition to summary statistics, the data points behind means, medians and variance measures should be available. If there are restrictions on publicly sharing data or code —e.g. participant privacy or use of data from a third party—those must be specified.

Reviewer #1: Yes

Reviewer #3: Yes

PLOS authors have the option to publish the peer review history of their article (what does this mean? ). If published, this will include your full peer review and any attached files.). If published, this will include your full peer review and any attached files.

**Do you want your identity to be public for this peer review?** For information about this choice, including consent withdrawal, please see our For information about this choice, including consent withdrawal, please see our Privacy Policy ..

Reviewer #1: **Yes:** Mingyue ZhengMingyue Zheng

Reviewer #3: No

---

## [Editor Report · Acceptance letter]

PCOMPBIOL-D-25-02332R1

Overcoming Extrapolation Challenges of Deep Learning by Incorporating Physics in Protein Sequence-Function Modeling

Dear Dr Chen,

I am pleased to inform you that your manuscript has been formally accepted for publication in PLOS Computational Biology. Your manuscript is now with our production department and you will be notified of the publication date in due course.

With kind regards,

Anita Estes
